# Custom Scanning Hyperspectral Imaging System for Biomedical Applications: Modeling, Benchmarking, and Specifications

**DOI:** 10.3390/s19071692

**Published:** 2019-04-09

**Authors:** José A. Gutiérrez-Gutiérrez, Arturo Pardo, Eusebio Real, José M. López-Higuera, Olga M. Conde

**Affiliations:** 1Photonics Engineering Group, Universidad de Cantabria, 39006 Santander, Cantabria, Spain; gutierrja@unican.es (J.A.G.-G.); eusebio.real@unican.es (E.R.); miguel.lopezhiguera@unican.es (J.M.L.-H.); 2Instituto de Investigación Sanitaria Valdecilla (IDIVAL), 39011 Santander, Cantabria, Spain; 3Biomedical Research Networking Center—Bioengineering, Biomaterials, and Nanomedicine (CIBER-BBN), Av. Monforte de Lemos, 3-5. Pabellón 11. Planta 0 28029 Madrid, Spain

**Keywords:** biomedical optical imaging, hyperspectral imaging, systems modeling, system implementation, system integration, benchmark testing

## Abstract

Prototyping hyperspectral imaging devices in current biomedical optics research requires taking into consideration various issues regarding optics, imaging, and instrumentation. In summary, an ideal imaging system should only be limited by exposure time, but there will be technological limitations (e.g., actuator delay and backlash, network delays, or embedded CPU speed) that should be considered, modeled, and optimized. This can be achieved by constructing a multiparametric model for the imaging system in question. The article describes a rotating-mirror scanning hyperspectral imaging device, its multiparametric model, as well as design and calibration protocols used to achieve its optimal performance. The main objective of the manuscript is to describe the device and review this imaging modality, while showcasing technical caveats, models and benchmarks, in an attempt to simplify and standardize specifications, as well as to incentivize prototyping similar future designs.

## 1. Introduction

In the past decade, the advent of embedded computing has produced a new array of imaging systems that, while based upon mature technologies, have and will increase in complexity. Multi- and hyperspectral imaging (MSI/HSI) is an imaging modality capable of obtaining spatially resolved spectral information of a subject of interest, and one of many areas of research that can also exploit these new developments. This technique has been thoroughly employed in remote sensing [1], crop analysis and agricultural and soil science [2,3,4], as well as food quality control [5,6,7]. In the field of Biomedical Optics, MSI/HSI systems are used in various diffuse optical imaging technologies, such as Spatial Frequency Domain Imaging (SFDI) [8], Single Shot Optical Properties (SSOP) [9], and corrected fluorescence imaging (qF-SSOP) [10], among others [11]. While handheld devices are becoming readily available on the market [12], for more specific applications such as SFDI, SSOP or qF-SSOP, having control over the optics and mechatronics of the system (e.g., automatic gain control, exposure control, communications with additional instrumentation, non-proprietary API control, etc.) is an unavoidable requirement. In these fields of research, then, it is common practice to design highly controllable, fully customizable, open-architecture systems with high spectral and spatial resolution.

The design modalities of HSI systems are well documented [13], and the multiplicity of different spectroscopic configurations for material classification is certainly notable, with various design characteristics and tradeoffs (see [14,15,16]), and increasing in number in recent years (e.g., [17,18,19,20,21,22]). The main characteristics of the most ubiquitous imaging systems are left in Table 1. In scanning imaging systems in particular, the spectra of a single point or line in an object plane is measured at a time, and a complete spatial image is achieved by moving either the sample, the imaging device, or components that may change the direction of acquisition (a thorough description of these devices can be found in the literature [12,13]). To the authors’ knowledge, unfortunately, there are no reported reviews on scanning imaging system design, modeling, and benchmarking, so relevant tradeoffs may have been left unexplored. Prototyping a scanning imaging system is indeed a challenging job but, as will be described in the following sections, in practice there are a few caveats and considerations which, once thoroughly reviewed, greatly illustrate the most relevant difficulties of HSI imaging system design. Encouraging the exploitation and implementation of more custom-built devices for biomedical applications, as well as hopefully endowing some degree of standardization for future cases, may be deemed desireable.

The main objective of this article, therefore, is to propose a set of metrics that represent scanning devices such that the key variables that define image quality become adequately established, since overall system performance is not only dependent on the properties of its constituent parts, but also on how efficiently these elements are combined together. A second objective is to describe a custom-built HSI prototype, which presents a structure that may constitute a novel approach, and to use these defined benchmarks to optimize its properties and evaluate where it could be improved. The proposed work pursues the optimization of these efficiency metrics, which could also be extrapolated to other HSI imaging modalities, comparing each other in terms of systems efficiency, evaluating imaging artifacts and acquisition delays, or using them with already existing custom multi- and hyperspectral scientific prototypes for debugging and design optimization.

The document is structured as follows. Section 2 describes a model proposition for a pushbroom/scanning system, as well as a general overview of a custom-built prototype and, finally, the proposed set of performance benchmarks for scanning devices. In Section 3, these benchmarks are tested on the prototype with a series of simple protocols. Finally, some optimization considerations and the complete characterization of the pushbroom scanning system are discussed in the last pages of the article.

## 2. Materials and Methods

### 2.1. Fundamental Parts of a Scanning Imaging System

Figure 1 shows a brief summary of the components that comprise a far-field scanning imaging device. Consider the following list as a generic description of the main components of any scanning system. Their names and properties will be the starting point for the evaluation of the custom device:**Optical subsystem.** An optical component will take a line in the field of view (FOV) and break it up into spatial and spectral information, which will be measured by a camera. There will likely be focusing optics, a spectrographic element and their corresponding couplings and interfaces. From a systems perspective, optics will constrain our *wavelength range of operation*, and the spectrograph and lens will limit the overall *spectral resolution*, i.e., the minimum spectral distance between two monochromatic signals that can be spatially separated.**Imaging device.** Light spatially separated by wavelength will reach a camera, which will capture incoming photons across its active pixel sensor. Different wavelengths will, then, be differentiated by their arrival onto different pixels. Consequently, several imaging parameters will depend on sensor characteristics, namely *quantum efficiency* (and wavelength range of operation), *exposure time*, *camera gain*, *pixel depth* and *spectral resolution*. While the first parameters will depend on the specific characteristics of the camera, the latter will be also related to the spectrograph’s ability to separate wavelengths. In the case of line-scanning hyperspectral devices, the camera will also specify the *spatial resolution* for one of the two spatial axes.**Actuator subsystem.** The actuator subsystem will usually constitute the sole moving part of the imaging device. In pushbroom systems, for example, either the imaging device or the platform where the sample is placed is moved by a large actuator. In the case of rotating mirror scanners, an actuator/motor will drive a mirror fixed to an otherwise free-rotating axle. Depending on the driving mechanism, (e.g., belts and/or gear mechanisms) there will be mechanical imperfections—such as *backlash*—as well as *delays* due to serial communications between the main computer and the actuator controller/driver. Consequently, the main variables at play will be the *repeatability* of the drive system (i.e., its ability to obtain the same image pixel-wise when repeating the same motion) and time delays due to serial communications. If using a stepper motor, its properties (e.g., steps per revolution, stepper resolution, and degree of microstepping) will define the actuator system and its ability to produce high-resolution images.**Communications and storage.** Image acquisition and actuation control must be governed by a main computer, which must communicate with both camera and actuator subsystems, as well as manage data storage and image calibration. Storage must also be sufficiently efficient so that storing acquired spectra does not impose a significant hindrance on the overall system. We will assume that the imaging system must be able to communicate with other instrumentation and computers within a laboratory network. Camera-computer communications will impose a restriction on the total time per measurement, and therefore *communication delays* as well as *driver and application program interface (API) delays* will play a significant role in our efficiency models.

### 2.2. Developed Rotating Mirror Scanning HSI Imaging System

The developed device is a custom-built rotating-mirror scanning HSI device with an embedded computer for storage and control, summarized in Figure 2 and Table 2, Table 3, Table 4 and Table 5. As a rotating mirror system, it has the same properties of a pushbroom system, while at the same time minimizing the number of moving parts, and thus actuator delays. It consists of an *ImSpector V10E* spectrograph (Specim, Spectral Imaging Ltd., Finland) paired with a CS-mount, 10 × 5 mm–50 mm f/1.3 varifocal objective lens with manual iris and zoom, originallly prepared for 1/3” cameras, (T10Z0513CS-3, Computar Optics Group, Cary, NC, USA) and a *Mako G-223B NIR* CMOS camera (Allied Vision Technologies GmbH., Stadtroda, Germany). Coupling the CS-mount objective lens onto the ImSpector and a 2.2 megapixel (2048 × 1088) C-mount camera provides a shorter minimum working distance and angle of view, at the expense of losing a few spectral pixels and half of the spatial pixels of the camera, resulting in a 1200 × 1200 px FOV. This ensemble allows the reading of a single hyperspectral line, which can then be swept across the sample with a rotating metal mirror system powered with a NEMA 17 stepper motor and a GT2 pulley array, achieving 19,200 steps per revolution with minimal backlash and CNC-grade (Computer Numerical Control) repeatability.

Communications are handled by a generic Gigabit Ethernet router, while DHCP services and camera control are handled by an *Odroid XU4* (Hardkernel Co., Ltd., GyeongGi, South Korea) running Ubuntu 16.04. Dedicated Python 3 software was prepared to handle pymba, a Python 3 wrapper for the Vimba 2.1.3 SDK. Camera control within the laboratory can be achieved through a simplified Python webserver running Flask. Gain control is in the 0–25 dB range, and camera exposure can be set anywhere in the 10 ms–2 s time range. Up to 1.4 MP (1200 × 1200 px) image resolution can be achieved if necessary, although smaller images can be stored via software. The number of sensor bins in the spectral axis is 1088 channels in total, but the total number of spectral channels will be given by the spectrograph. Camera aperture, focus and zoom can be controlled manually.

### 2.3. Efficiency Benchmarks and Calibration Procedures

We shall obtain a set of expressions that can serve as adequate overall system performance metrics, as well as calibration and correction protocols for spatial and spectral information. While the former will provide insight into this particular imaging modality, the latter are standard procedure in hyperspectral imaging.

#### 2.3.1. Lines per Second (LPS)

Analogous to frames per second (FPS) in conventional systems, it is the inverse of the total time required to measure a single line, namely the exposure time plus additional time delays:(1)$$\mathrm{LPS}=\frac{1}{T_{\mathrm{line}}}=\frac{1}{T_{\exp}+\underset{T_{\mathrm{extra}}}{\underset{︸}{T_{\mathrm{rx}}+T_{\mathrm{api}}+T_{\mathrm{mcpy}}+T_{\mathrm{act}}}}+T_{\mathrm{rand}}}.$$

In this equation, *T*_line_ represents the total time needed for acquiring a single line, and it is constituted by a combination of various different time delays and intervals that take place during acquisition:Exposure time, denoted by *T*_exp_, is the time that a sensor will spend capturing photons from a single line of the FOV (shown in Figure 1). Ideally, image acquisition should only consist of integer multiples of a preset exposure time, and image storage and communications should be immediate. It will be shown that this is not the case.Transfer/receiving time *T*_rx_ is the relationship between file size and network speed within the device. Transfer time over an Ethernet network will be practically constant and, thus, for a fixed image size, transfer time between the camera and the main computer will be fixed. As a general rule, the total transfer time will be given by
(2)$$T_{\mathrm{rx}}=\frac{h\cdot w\cdot b}{R_{b}}\cdot \Omega ,$$
where *h*, *w* are the native height and width (in pixels) of the camera, *b* is the bit depth (either 8 or 12 bits), *R_b_* is the bitrate (in bits/s), and Ω is the overhead of the protocol. In our practical case, where Vimba uses UDP (with an overhead of 1.9%, Ω = 1.019) on a 2048 × 1088 pixel camera with 8 (or 12) bits per pixel connected to a Gigabit Ethernet router, we obtain 18.16 ms and 27.24 ms of transmission time for 8- and 12-bit images, respectively. This will introduce a limitation of about 55.06 and 27.04 frames per second, respectively, since the camera will be the only device sending packets to the main computer, and switching time will be considered negligible.Camera Application Program Interface (API) delays, *T*_api_. Any imaging sensor will be controlled by the computer via an API or driver library. Driver and API delays should be constant delays that an imaging device presents that can be shown to be due to runtime execution of precompiled API dynamic libraries.Memory buffer copying and transfer times *T*_mcpy_. They represent buffer copying delays due to internal bus limitations within the main computer, i.e., between its network interface and memory.Actuator/mirror movement delays *T*_act_. These comprise delays in serial communications between the main computer and the actuator subsystem, and due to stepper motor movement, which will be negligible for high-resolution images, but noticeable in low-resolution acquisition.There will be additional, random delays, mostly due to Operating System (OS) interruptions. We will denote them as *T*_rand_. Unless we work with a real-time operating system, there shall be random interruptions during acquisition, due to the OS scheduler setting the measurement API to background due to priority issues. This phenomenon has a stochastic nature and cannot be controlled unless a real-time operating system is used.

For our model, *T*_extra_ = *T*_rx_ + *T*_api_ + *T*_mcpy_ + *T*_act_ represents all the additional delays in the system that are not random. This value will be estimated as a constant during nonlinear least squares fitting in the following sections.

#### 2.3.2. System Efficiency

The *efficiency* of a scanning imaging device can be described as the ratio between the specified exposure time per sensor capture and the actual time per line:(3)$$\epsilon =\frac{T_{\exp}}{T_{\mathrm{line}}}=\frac{T_{\exp}}{T_{\exp}+T_{\mathrm{extra}}+T_{\mathrm{rand}}},$$
and this measurement will always be *ϵ* ≤ 1, since
(4)$$\epsilon_{\max}=\lim_{T_{\exp}\to \infty }\left\{\epsilon \right\}=1.$$

#### 2.3.3. Object Plane Curvature

Using a rotating mirror for scaning imposes a curvature distortion on the FOV of the camera, such that the region of space that remains focused is cilindrical, not planar. This nonlinear transformation can be inferred theoretically by considering Figure 3. Angle *θ* is the orientation angle of the mirror with respect to the normal of the object plane, and *γ* represents the angle between the object plane normal and the incoming light ray. By the definition of the tangent function, we can infer that the line position along the object plane, *y_d_*, is given by:(5)$$y_{d}=r\cdot \tan\left(\gamma \right)=r\cdot \tan\left(\frac{\pi }{2}-2\theta \right),$$
and the difference between the normal distance to the object plane and the curved object plane, Δ*r* = *r* − *d*, with $r=\bar{AP}$ and $d=\bar{AP^{\prime }}$, is equal to
(6)$$\Delta r=r\left(\sqrt{1+\frac{1}{\tan^{2}\left(2\theta \right)}}-1\right).$$

Additionally, the width of objects in a line (*x*, *θ*) will be linearly distorted by a scaling factor, which is also a nonlinear function of mirror angle *θ*:(7)$$h\left(\theta \right)=\sqrt{1+\frac{1}{\tan^{2}\left(2\theta \right)}}$$

Therefore, every pixel in the image (*x*, *y*) will correspond to a spatial position (*x_d_*, *y_d_*), where
(8)$$\begin{array}{cc} y_{d} & =r\cdot \tan\left(\frac{\pi }{2}-2\theta_{k}\right), \end{array}$$
(9)$$\begin{array}{cc} x_{d} & =x\cdot \Delta x\cdot h(\theta_{k}), \end{array}$$
(10)$$\begin{array}{cc} \theta_{k} & =\Delta \theta \cdot (y-y_{\pi /4})+\pi /4, \end{array}$$
where Δ*x* is the spatial resolution of the measured line in millimeters (which will be generally a function of lens optical angle of acceptance and *r*), *y*_*π*/4_ corresponds to the pixel index where the ray path is fully normal to the object plane (also, where *θ* = *π*/4), and the angular resolution of the stepper motor Δ*θ*, in our case, will be the total number of steps per revolution times a stepping factor *s*:(11)$$\Delta \theta =s\frac{2\pi }{N_{res}}=s\frac{2\pi }{19,200}.$$

A representation of the complete nonlinear transformation (without defocusing) is illustrated in Figure 4. As we diverge from the linear range of operation, the captured image becomes deformed and downscaled, since we are receiving rays from points that are further away from our object plane normal.

For this approach, the tangent function can be approximated for small values of *θ* as a linear transformation between mirror angle *θ* and position *y_d_* in the field of view within focus. In particular, for values γ∈Γ=[−π/8,+π/8] (θ∈Θ=[π/4−π/16,π/4+π/16]), the approximation
(12)$$\begin{array}{cc} y_{d} & \approx {\hat{y}}_{d}=-r\left(2\theta -\frac{\pi }{2}\right), \end{array}$$
(13)$$\begin{array}{cc} x_{d} & \approx x\cdot \Delta x \end{array}$$
can hold for Δ*r*/*r* ≤ 8% (smaller errors can be obtained by reducing this range further). If lens aperture is kept within a range such that optical depth of field remains over Δ*r*, then a clear image can be captured. Any system should be verified in a real context (at least for a particular value of *r*), so that its linear range can be specified.

#### 2.3.4. Microstepping Accuracy

Once the main distortions due to using a rotating mirror have been considered, there will be additional accuracy noise caused by the remaining mechanical factors, namely backlash and microstepping. We will assume that any remaining errors will be coming from random sources and are Gaussian-distributed, i.e., $n∼N(0,\sigma^{2})$, hence simplifying the remaining uncertainties of the model. The standard deviation of these random events will be estimated by histogram approximation of a Gaussian probability density function (PDF), and is expected to be negligible for high-resolution images and quite relevant in low-resolution acquisition, since motor inertia will correct current variations in the windings of the stepper [24].

#### 2.3.5. Spatial Resolution

Spatial resolution is typically estimated using a resolution test chart. The one we choose to use is a 1951 USAF Resolution Test Chart (MIL-STD-150), printed with reference lines with known lengths. Such lines provide a relationship between pixel positions and actual distances in the image plane.

#### 2.3.6. Spectral Calibration

Wavelength separation through a spectrograph is known to be non-linear in nature. Generally, since a given spectrograph has a fixed response over time, spectral characterization only needs to be performed once. There are two steps required in order to fully calibrate a hyperspectral system. First, the polynomic response of the spectrograph as a function of pixel position must be found. The characteristic polynomial of a spectrograph, *λ* = *P*(*p*), is a least-squares fit that attempts to relate a spectral wavelength *λ* to its pixel position *p* within the sensor:(14)$$\lambda =a_{0}+a_{1}p+a_{2}p^{2}+\cdots +a_{d}p^{d}.$$

Having a material with well-known reflectance peaks, a polynomial of degree *d* can be obtained through least-squares regression with *d* or more reference peaks, *λ*_0_, ⋯, *λ_d_*, which can be then identified at specific pixel locations, *p*_0_, ⋯, *p_d_* [25].

After being able to relate pixels with wavelengths, we can then establish the *spectral resolution* of the device. This was achieved via a CAL-2000 Mercury–Argon light source (Ocean Optics Mikropack, Ostfildern, Germany) oriented towards the rotating mirror, which shall be kept at a 45° angle. After background substraction, the spectral emission lines of both gasses should be visible. Since their spectral emission lines are well-known, finding the smallest discernible lines will provide a lower bound for spectral resolution.

#### 2.3.7. Dark Image Substraction

The term *hot pixels* refers to the appearance of bright pixels at long exposure times due to sensor lattice damage; its correction can be achieved via *dark image substraction*, quite common procedure in astronomical imaging calibration [26]. Hot pixels in the sensor must be corrected by characterizing their behavior as a function of exposure time when they are not exposed to any light. This can be achieved by closing the aperture completely and obtaining several measurements. By obtaining the baseline pixel values for several exposure times, hot pixel background signals can be modeled in each pixel by
(15)$$h_{bg,ij}=a_{ij}T_{\exp}+b_{ij},$$
where indices *i*, *j* indicate the pixel’s position in the sensor, *a_ij_* and *b_ij_* are the coefficients that model the linear behavior of the pixel as a function of exposure time *T*_exp_, and *h_bg,ij_* will be the estimated hot pixel value that should be subtracted to each pixel for every new measurement. This relationship should be occasionally verified, in order to keep track of the number of damaged pixels in the sensor [27].

#### 2.3.8. Light Source Stability and SNR

Without properly characterizing the main source of optical power, we may incur in inaccuracies in power output and, therefore, in reflectance measurements. Ideally, for biomedical applications it is preferable to use stable and power-controlled devices, such as monochromatic lasers and/or supercontinuum light sources. As a preliminary approach, it is sufficient to use a tungsten halogen lamp in the Vis–NIR range, due mainly to its thermal inertia, which provides light source stability in the millisecond-to-second scale. We shall procure two protocols for analyzing how the signal-to-noise ratio (SNR) of the camera is a function of exposure time and light source optical power.

We will use either one or two 1-kW tungsten halogen bulbs for all measurements, depending on the absorption of the sample. For the first experiment, we turn on the light source and take a low resolution snapshot of an illuminated white Spectralon calibration reference. This is repeated several times (20 times in our case), and each snapshot is timestamped and stored. The average Spectralon reflectance value for each timestamp will be calculated, and then plotted as a function of time. If thermal inertia works well under standard temperature conditions, the average value of this Spectralon reference should become constant over time.

The second experiment assumes that the received sensor counts are a function of backscattered optical power. Therefore, the electrical current transformed into a byte in each pixel will be consequence of the sum of two powers:(16)$$P_{rx}=P_{s}+P_{n},\;P_{n}∼N\left(0,\sigma^{2}\right),$$
where *P_rx_* is the received optical power on that pixel, *P_s_* is the optical power coming from the sample, and *P_n_* is Additive White Gaussian Noise (AWGN). Under this assumption, the average SNR as a function of wavelength and exposure time can be calculated via the following expected value ratio:(17)$$SNR\left(T_{\exp};\lambda \right)=10\log_{10}\left(\frac{P_{s}}{P_{n}}\right)=10\log_{10}\left(\frac{E[P_{rx}\left(T_{\exp};\lambda \right)]}{E[{\left(P_{rx}\left(T_{\exp};\lambda \right)-E[P_{rx}\left(T_{\exp};\lambda \right)\right)}^{2}]}\right),$$
which for the given assumption is equivalent to calculating
(18)$$\begin{array}{cc} \hat{\mu }\left(T_{\exp};\lambda \right) & =\frac{1}{N}\sum_{n=1}^{N}x_{k}\left(T_{\exp};\lambda \right), \end{array}$$
(19)$$\begin{array}{cc} {\hat{\sigma }}^{2}\left(T_{\exp};\lambda \right) & =\frac{1}{N}\sum_{n=1}^{N}{\left(x_{k}\left(T_{\exp};\lambda \right)-\hat{\mu }\left(T_{\exp};\lambda \right)\right)}^{2}, \end{array}$$
(20)$$\begin{array}{cc} SNR(T_{\exp};\lambda ) & =10\log_{10}\left(\frac{\hat{\mu }}{{\hat{\sigma }}^{2}}\right), \end{array}$$
where here *x*_1_, ⋯, *x_N_*(*T*_exp_; *λ*) are several pixels of a reference material measured at a given exposure time and wavelength, $\hat{\mu }$ is a maximum likelihood (ML) estimate of the average reflectance of such material, and $\hat{\sigma^{2}}$ the ML estimate of the variance of the light source. These calculations should be performed on reference-calibrated reflectance data, so two Spectralon captures are required for each evaluated exposure time, and square-law losses due to light source directionality can be corrected.

## 3. Results and Discussion

Once all constraints and properties of our model are adequately defined, we must characterize the imaging system and verify that, indeed, it behaves as described. These results have been obtained from the device explained in Section 2.2, and attempt to serve as an example of how to measure whether or not each component is being exploited to its maximum potential. Here, we shall focus on the elements of the reviewed model that have not been studied by previous work: (1) timing issues, (2) spatial distortions, (3) microstepping noise, and (4) overall efficiency.

### 3.1. Timing and Delays

Measuring time spent in each of the modeled operations is an adequate first step in characterizing system efficiency. In Figure 5, the total time per line is partitioned between its essential tasks. For this experiment, each measured line is separated by a single motor step (i.e., the device is working at full resolution). This was tested at 8-bit and 12-bit depth, since bit depth will influence the total time spent transmitting data. As expected, with no other processes running in the system, transfer and buffer copying only take a small fraction of the total time, while a fixed time is spent communicating with the camera and the rest of the time is spent acquiring light.

By obtaining the inverse of this time per line, we can represent the rate of acquired lines per second of the scanning device. This calculation is shown in Figure 6, where the dots represent the inverse of the bars in Figure 5, and the continuous line is a nonlinear least squares fit to 1/(*T*_exp_ + *T*_extra_), being *T*_extra_ the constant time parameter that we wish to estimate. Additionally, the ideal model, i.e., 1/*T*_exp_, is shown as a dotted black line. The different graphs represent the average lines per second when parts of the system are disabled: the red plot represents normal acquisition, while in the case of the blue line the rotating mirror remains static (i.e., no serial commands are sent to the rotating mirror actuator) and, for the green line, only acquisition commands are being executed, and acquired images are discarded (not stored into memory). Driver and API delays are, in theory, constant values that can be shown to be due to runtime execution of precompiled dynamic libraries. Nevertheless, random variations in timing are clearly visible after adjusting for our model, and are related to OS schduler interruptions and multithreading management, since acquisition, actuator control, storage and RAM management are run by different processes. In order to obtain accurate, stable LPS measurements, we must use application-specific circuits, and/or real-time operating systems.

Additionally, it must be noted how, regardless of exposure time, all measurements are limited by the maximum FPS rate that the camera and/or network protocol can manage (49.5 FPS/LPS for 8 bits is limited by camera performance, 27 FPS/LPS for 12 bits is limited by network speed). All measurements are bounded from above from the value given by Equation (Equation 2) due to network bandwidth, overheads in transmission, as well as transmission delays, router switching delays, and DLL runtime delays (which cannot be optimized since the Vimba API is already compiled and no source code is available). These issues produce an additional limit, as well as random acquisition speed variability. Some clinical approaches, like fluorescence quantification, will require 12-bit images, while other methods could benefit of much faster acquisition times rather than a higher pixel resolution; these issues should be considered in advance depending on the application.

A similar approach can be done for the other modalities shown in Table 1, considering their ideal efficiency or throughput (Table 6). For the spatial and spectral resolution considered in Table 1, a rotating mirror scanning system would score favorably with respect to other modalities, considering that rotating a small mirror will take a shorter amount of time and less power consumption than displacing either the sample or the imaging system (i.e., as indicated in the table, *T*_act_ ≪ *T*_move_).

### 3.2. Image Distortions and Noise

Since focusing optics are independent from the actuator subsystem in many cases, the object plane will inevitably suffer nonlinearities. A rotating mirror with a stepper motor will incur three types of measurement errors. First, the object plane will be curved, as the focusing distance will be fixed for any mirror angle; this curvature effect must be limited to a linear range, and its effects must be quantified. Second, there will be backlash in the transmission system that connects the motor to the mirror axle; such backlash can damage repeatability, and must be minimized. Third, and finally, there will be microstepping noise due to spurious currents in both driver and stepper rotor; those must be minimized by using an adequate stepper driver and microstepping factor.

Although we can model theoretical distortion due to mirror rotation and avoid exceeding its linear range of operation, overall distortion and noise can (and should) be characterized by imaging an Amsler grid with various lengths and forcing the rotating mirror to capture a half/quarter rotation at maximum resolution. Then, the first finite difference along the *y* axis can be obtained, which allows for edge detection in the grid. The variation in grid cell width can be approximated by the nonlinear scaling factor of Section 2.3.3, and therefore a nonlinear least-squares fit of Equation (Equation 5) can be obtained. This deterministic distortion allows for obtaining *r*, the total distance to our object, as well as isolating any remaining errors coming from other sources. The result of this calculation is left in Figure 7. To obtain a successful alignment, an Amsler grid with 6 × 6 mm squares was printed at maximum resolution, screwed to an optical table, where a SMS20 Heavy Duty Boom stand/crane (Diagnostic Instruments, Inc., Michigan, US) was situated holding the instrument above the grid. Then, horizontal (*x*-wise) alignment between grid and camera was achieved by constantly previewing the image and overlaying a series of horizontal lines in a custom-designed graphical user interface (GUI), moving and fixing the stand until centering and alignment of the center dot of the grid was achieved at *θ* = *π*/4 (or, alternatively, *γ* = 0). The relationship between position *y* and angle *θ* is linear within the range ±*π*/8 ≈ ±0.3926 and distortions in the *x* axis can be corrected by an inverse affine transformation.

The residual in the right subplot shows a cosine-like pattern that is likely due to misalignment errors that cannot be compensated by the GUI and manual adjustment. While this non-random variation cannot be corrected any further without more precise equipment, random variations in *y_d_* can still be accounted for by FFT clipping this signal down to its average noise spectral power density. The results of this experiment are shown in Figure 8, where the left plot shows the variations in grid size beyond misalignment, and the right subplot presents a histogram and a Gaussian fit for an estimated mean *μ* and variance *σ*^2^. This variability is therefore in the order of *σ* ≈ 0.2 mm at a distance of 348.73 mm and a FOV size of 120 × 120 mm, which can be considered negligible for far-field measurement.

### 3.3. Spectral Calibration

As specified in the Materials and Methods section, a Wavelength Calibration Standard plate with tabulated absorption peaks was imaged with a tungsten halogen lamp. The result of finding at least 8 tabulated peaks is displayed in Figure 9, where absorption peak distribution before and after calibration can be seen. Spectral resolution was empirically obtained using a calibration light source with known proximal emission peaks within the wavelength range and peak distances within the range tolerated by the spectrograph. The top subfigure in Figure 10 shows such measurement, with superimposed well-known argon and mercury peak wavelengths. Lines closer than 3 nm to each other were not discernible (e.g., 576.96 and 579.07 nm), which means that spectral resolution shall be bounded by 3 nm. This provides an upper bound on the number of spectral channels, namely 218 (i.e., 5 pixels or 3 nm per channel). Taking resolution into consideration, the spectrograph response can be calculated, resulting in *λ*(*p*) = 370.79 + 2.68*p* − 0.0001*p*^2^, with *p* being one of the 218 channels. This fairly linear response is expected.

### 3.4. Spatial Resolution

A spatially calibrated image of the aforementioned resolution test chart is displayed in Figure 11. Closeups for 2, 1, and 0.5 millimeter-wide bands are provided on the right side of the figure. Vertical resolution (in red) degrades much faster than horizontal resolution (in green), with horizontal lines (vertical resolution, in red) blurring out at about 0.5 mm, and vertical lines (discerned by horizontal resolution, in green) can be distinguished at up to 0.2–0.3 mm. If spatial resolution is defined as the maximum between horizontal and vertical resolution, then the spatial resolution of the imaging device is 0.5 mm with the camera placed at a 35 cm distance from the test chart. Given the actuator noise shown in Figure 8, we can conclude that vertical resolution is currently limited by microstepping noise.

### 3.5. Light Source Calibration and SNR

The outcomes of the experiments described in Section 2.3.8 are shown in Figure 12. The left subplot provides empirical support to explain how the SNR of reflectance signals behaves as a combination of exposure time, reference reflectance, and camera quantum efficiency. The right subplot shows the average received optical power during the first 20 seconds of operation following the activation of the halogen light source. From these results, we can conclude that (1) the response of the camera is linear with respect to exposure time, (2) that the SNR of reflectance data is limited by the spectral response of the optical imaging equipment, and (3) that our light source is stable, on average, within the timescale of operation.

### 3.6. Benchmark Results and Characterized Equipment

Once our image is as clear as possible, spectral information has been calibrated, and timing issues are well characterized, it is time to benchmark overall system performance. While LPS values show the maximum acquisition speed of the system, efficiency values (as described in Section 2) will show how much the maximum LPS rates are throttled by time not spent in acquiring lines. For Figure 13, the same LPS measurements are repeated and efficiency is plotted (scatter plots) and nonlinear least-squares fit to Equation (Equation 3) (smooth line plots), for the three scenarios explained in Section 3.1 and for 8- and 12-bit images. The behavior of this graph is expected, as efficiency is a monotonically increasing function with respect to exposure time, and rises quickly as less tasks take control of the main computer and spend time performing non-essential tasks.

Although efficiency can be seen as an indirect measurement of LPS rates, it can be considered an explicit benchmark in and of itself, since it only indicates the divergence between an ideal, theoretical measuring machine, and the device under test. It completely ignores every other specification of the system, and focuses on how much CPU time is dedicated to measuring only. Better cameras can be substituted in the system and its efficiency can be reevaluated, which facilitates device adaptation and updating to different acquisition environments. For example, higher efficiencies could be more practical in constantly moving environments (such as real-time, in-vivo diagnostics during a surgical procedure), while lower efficiencies can be sufficient for slower scenarios (i.e., ex-vivo imaging). Also, they could be used in industrial manufacturing as a way to quantitatively compare camera efficiency the various tradeoffs between resolution, gain, sensor quantum efficiency, network interface, and spectral range of operation, when selecting a sensor in a commercial device or process.

Another interesting conclusion is related to scanning hyperspectral devices as an imaging modality. Artifacts due to subject motion and/or metabolic changes, such as sample oxidation, changes in oxygenation, degradation, or decay will be more severe as the total time spent capturing is increased. While under these conditions other imaging modalities can prove more useful, a fair tradeoff between speed and resolution can be achieved in these systems, since the total time spent in an image with *N*_lines_ lines measured will be *T*_img_ = *N*_lines_*T*_line_/*ϵ*_avg_(*T*_exp_), where *ϵ*_avg_(*T*_exp_) is the average measurement efficiency for a given exposure time. Consequently, the higher the efficiency, the smaller the number of uncorrectable artifacts that could be found in an image with an adequate choice of resolution and exposure time.

Finally, by assuming that the model takes into account all timing and nonlinearity issues, its final characteristics can be summarized, as in Table 7. Understandably, some of its specifications (e.g., gain and exposure time ranges) are extracted from their parts’ corresponding tables, while the rest are a consequence of the benchmarks described throughout the article. Resolved distances, nonlinearity corrections, delays and efficiencies may be illuminating, for example, when comparing tradeoffs and similarities with known equipment, commercial or otherwise; both device manufacturers and applied optics scientists can compare, select, replicate and improve their systems, based on how well different parts perform together.

As for now, we can report on the protyping of a high-resolution, highly efficient hyperspectral imaging system, capable of 1.44 Mpx spatial resolution, 200 wavelength channels over the Vis–NIR range, fully modifiable optics, variable gain, exposure time and acquisition mode for HSI, SFDI, and SSOP measurements (as well as any additional experimental methods) which can be controlled via a RESTful Web Service from any computer within the laboratory network. An HSI image of a hand obtained with this device (*r* = 350 mm, *T*_exp_ = 200 ms, full resolution), as well as some random spectra within the image, can be seen in Figure 14. Color reconstruction is achieved via CIE 1931 Color Matching Functions (CMFs) applied on the acquired spectra, providing XYZ/RGB images with identical resolution. A near-infrared channel (Figure 14b), namely 803 nm, reveals existing vascular structures in the hand. Some random spectra are also displayed at the bottom plot, with 1% transparency for each spectrum, to stand out repeated features (i.e., skin and background spectra) from unlabeled spectra.

## 4. Conclusions

Prototyping hyperspectral scanning equipment is a multiparameter, interdisciplinary task that requires harmonious and efficient behavior amongst components at play. This manuscript has described and summarized the main obstacles faced when constructing a scanning HSI device and has defined a few ideal objectives (and benchmarks to measure them) which system developers can consider when building devices as described in Section 2.

As reviewed throughout the article, an ideal system will always spend the vast majority of its time performing acquisition, while minimizing all preprocessing, communications, storage and API-based delays as much as possible. In real case scenarios, full control of its specifications (gain, exposure time, focusing distance, aperture, zoom) within a local network is also desireable, especially with multi-purpose camera systems that unlock numerous potential biomedical imaging applications.

In order to determine how well a specific custom device behaves (and how much it can be improved), methods like measuring timing constraints and metrics such as system efficiency can provide information on how close to ideal, asymptotic behavior the system under test is. Optics-related parameters, such as nonlinear distortions and microstepping noise, have been characterized as well (at least for the case of rotating mirror scanning systems), and in most cases system redesign has been sufficient to keep them as minimized as possible. Some interesting results include the fact that API communications can be the most significant cause of timing innefficiencies in a scanning system, that spatial distortions can be almost completely eliminated by keeping the FOV of the camera within a linear range, and that vertical resolution is limited by microstepping noise.

Interesting future lines of research include, among others: (1) applying flatter, more stable light sources, such as supercontinuum laser sources, in order to achieve a flatter SNR with respect to wavelength; (2) using curved mirrors for depth-of-field and focus plane correction; (3) approaching other wavelength ranges, such as NIR-SWIR; (4) using Application-Specific Integrated Circuits (ASICs), Field Programmable Gate Arrays (FPGAs), or real-time operating systems for minimizing all the aforementioned delays; (5) using various spectrograph slit sizes and combinations with other focusing optics, such as macro and/or telecentric lenses; (6) evaluating sensor lattice damage with more complex hot-pixel models, since these exhibit slight nonlinear behavior; (7) studying the exact influence of object motion artifacts in this imaging modality; and (8) using profilometry for source intensity correction, especially when using far-field sources, such as structured light projectors. All these possible improvements stem from the evaluation of the described equipment via the proposed methodology.

A more general conclusion that can be reached from these experiments is that a full characterization is necessary when it comes to understanding how the different components of an imaging system relate to each other, so that the root causes of imperfections and/or slowdowns in image acquisition can be adequately detected, and design can be improved in further versions, especially when building and testing devices with clinical and surgical applications, where timeliness and precision are fundamental.

## Figures and Tables

**Figure 1 sensors-19-01692-f001:**
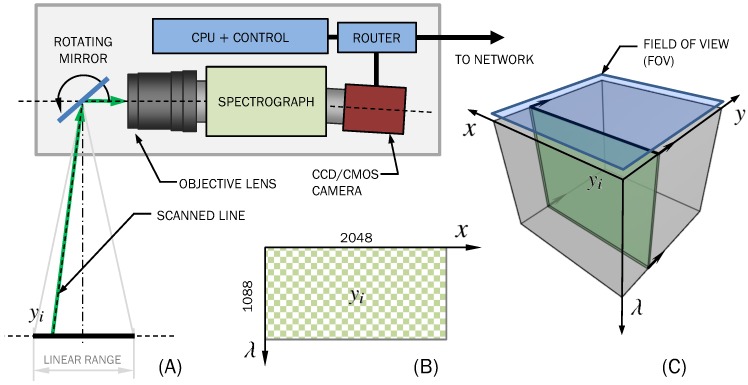
A generalized description of an external rotating mirror scanning imaging system. (**A**) A single focused line within the field of view can be captured at a time; (**B**) each line will consist of a two-dimensional sensor measurement, where the first axis will represent spatial dimension *x*, and the second one specifies wavelength, *λ*. (**C**) A complete hyperspectral image is obtained by moving the rotating mirror and storing more images for each of the mirror’s positions (*y*).

**Figure 2 sensors-19-01692-f002:**
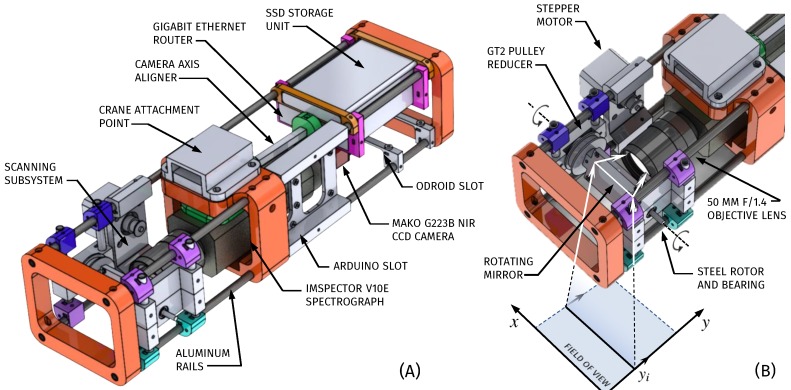
General schematic of the imaging prototype. (**A**) Overview and main components. (**B**) Closeup of the scanning mirror system. Spectra from a single line is read in a single shot, and then the scanning mirror allows for the capture of a full Field of View (FOV).

**Figure 3 sensors-19-01692-f003:**
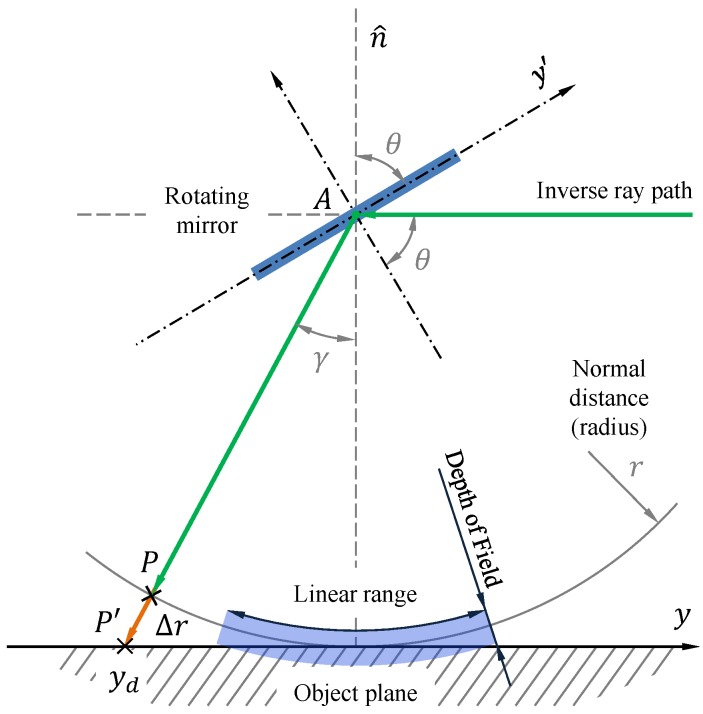
Rotating mirror geometry and linear range of operation. An image is acquired by varying rotating mirror angle *θ*. The optical path is increased by Δ*r*(*θ*), which produces distortion and defocusing. These artifacts can be considered negligible for an angle interval Θ (linear range).

**Figure 4 sensors-19-01692-f004:**
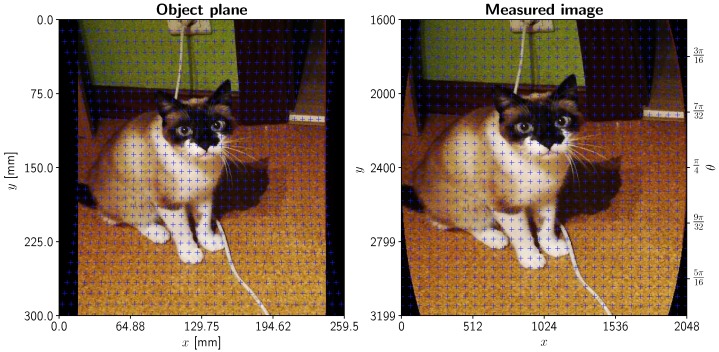
Image distortion in a scanning imaging system with a rotating mirror for a system height of *r* = 225 mm. Left plot: object plane, normal to the imaging system (axes in mm). Blue dots represent the positions of a subset of captured pixels that will form a uniform pixel grid in the measured image (right plot). Aspect ratio is not preserved so that distortions can be better highlighted.

**Figure 5 sensors-19-01692-f005:**
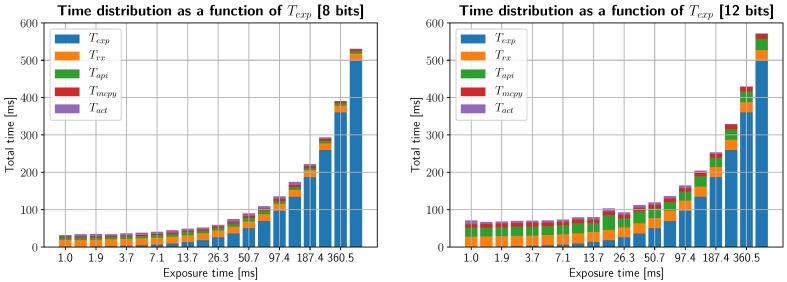
Line measurement time distribution for different exposure times, and bit depth (left and right subplots correspond to 8- and 12-bit depth, respectively). The total time spent in each line is a combination of exposure time *T*_exp_ (in blue), network transmission time *T*_rx_, dynamic library time *T*_api_ (in green), memory buffer copying *T*_mcpy_ (in red) and actuator delays *T*_act_ (in purple).

**Figure 6 sensors-19-01692-f006:**
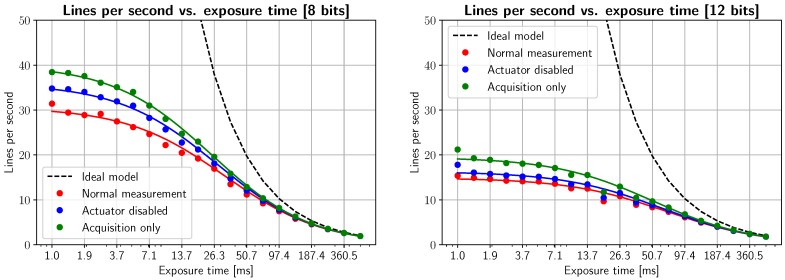
Acquisition speed (in lines per second) as a function of exposure time, disabling different parts of the imaging equipment. Measurements correspond to the colored scatter plots, while a nonlinear least-squares fit is superimposed as a continuous line plot for each scenario. As expected, disabling the scanning system and ignoring buffer copying allows us to reach the theoretical limit of the camera (40–50 FPS).

**Figure 7 sensors-19-01692-f007:**
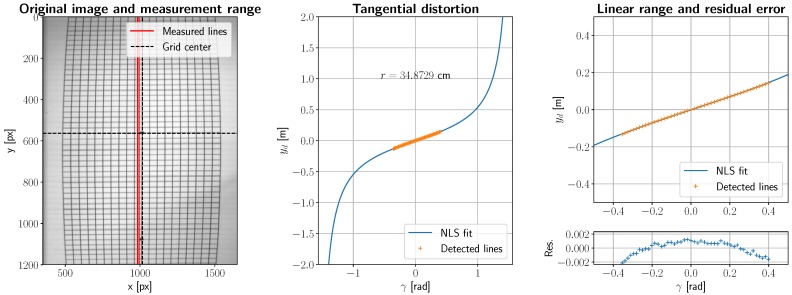
Tangential image distortions due to the selected rotating mirror configuration.

**Figure 8 sensors-19-01692-f008:**
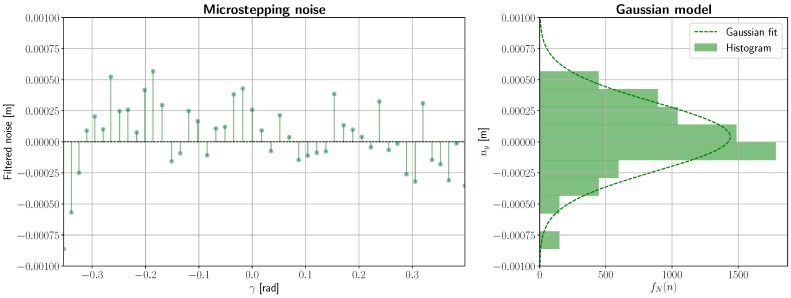
Image noise due to microstepping.

**Figure 9 sensors-19-01692-f009:**
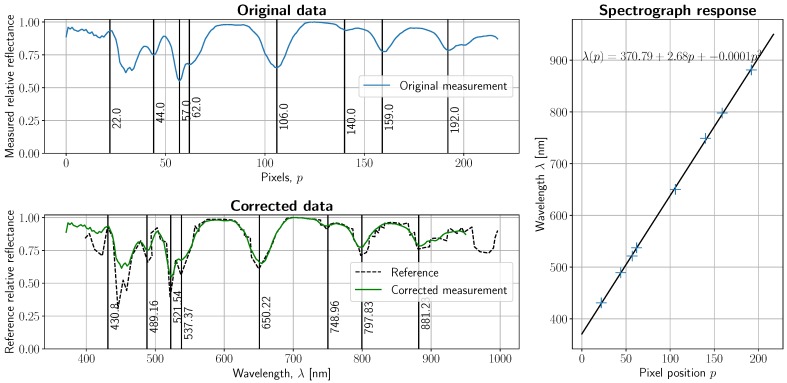
Spectral calibration with Wavelength Calibration Standard (WCS) WCS-MC-020. Top left: labeled absorption peaks in the measured reference. Bottom left: reference and measured spectra after calibration. Right: fitted quadratic polynomial with respect to labeled points.

**Figure 10 sensors-19-01692-f010:**
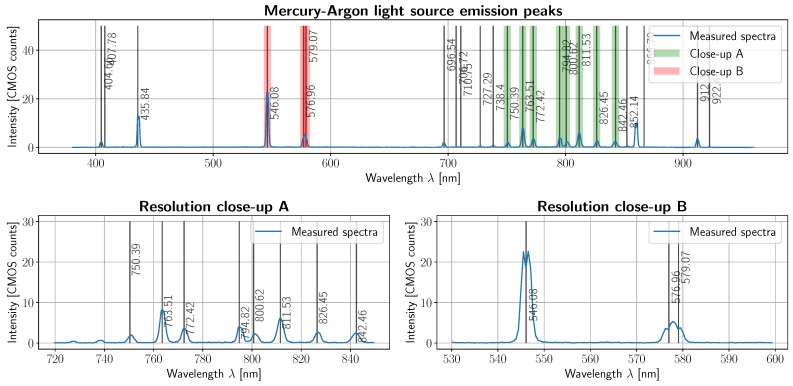
Spectral resolution bound obtained with the emission lines of a mercury-argon light source. Top subplot: labeled emission peaks and measured spectrum. Bottom left: closeup of argon emission peaks in the 720–840 nm range. Bottom right: closeup of some mercury emission peaks in the 530–600 nm range. Spectral resolution is bounded by 3 nm.

**Figure 11 sensors-19-01692-f011:**
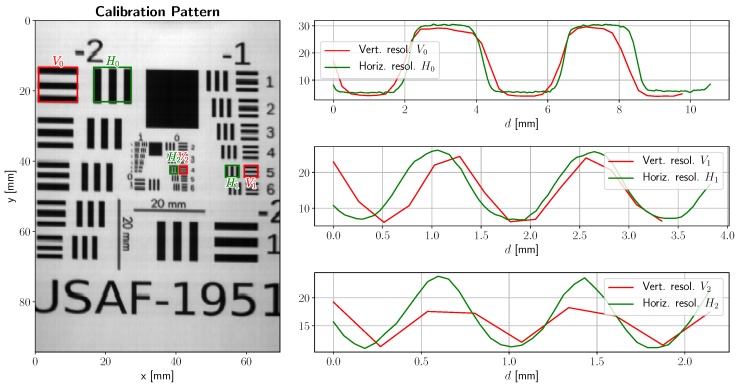
1951 USAF resolution test chart, average reflectance (left) and cross-section intensities at different band widths (right). Total resolution is limited by vertical (actuator) resolution, which could be improved in future versions by improving mechanical resolution and actuator repeatability. Best viewed in color.

**Figure 12 sensors-19-01692-f012:**
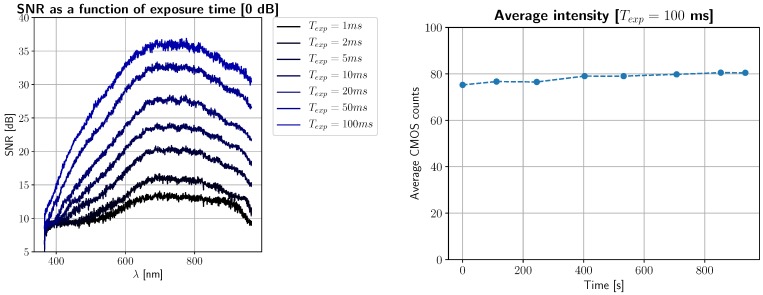
Left: Signal-to-noise ratio (SNR) of reflectance as a function of exposure time and wavelength. Right: Light source stability during the first 800 s of operation.

**Figure 13 sensors-19-01692-f013:**
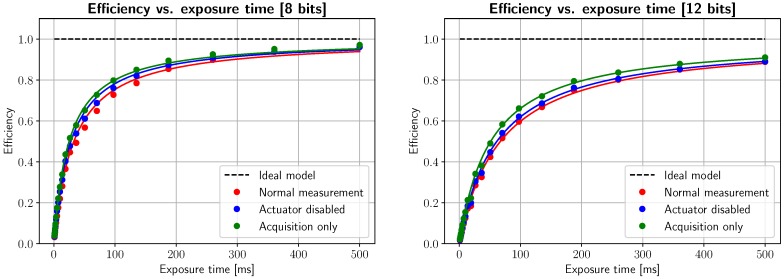
Measured system efficiency and nonlinear least squares fit, as defined by Equation (Equation 3) for 8- and 12-bit depth. As exposure time increases, the fraction of time spent in camera communications becomes negligible and efficiency increases. This is also increased with bit depth, as the data must traverse a bandwidth-limited network. As can be seen, mirror rotation and memory buffering produces a noticeable decrease in efficiency of about 10% for typical exposure times (100–300 ms).

**Figure 14 sensors-19-01692-f014:**
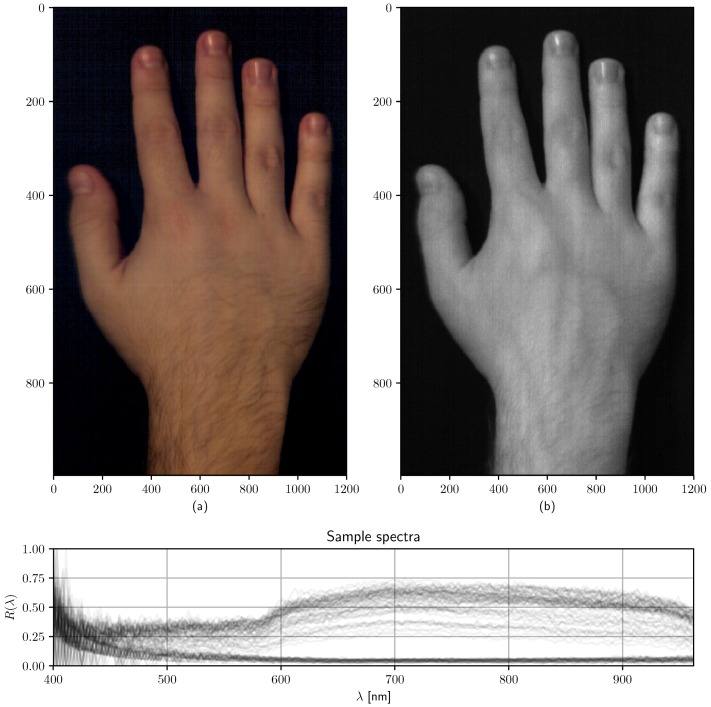
HSI system performance example. Top row: (**a**) RGB color reconstruction of a captured hand over the complete VISNIR range (400–1000 nm). (**b**) 803 nm channel image. Bottom row: 100 overlaid random spectra with 1/100 transparency, showing common spectral signatures.

**Table 1 sensors-19-01692-t001:** Frequent far-field reflectance hyperspectral imaging systems for biomedical applications, with its properties and applications (summary from [13,23]). Most devices trade off spatial, spectral resolution, and speed, depending on the configuration.

Imaging Modality	Spatial Channels/Resolution	Spectral Channels/Resolution
Snapshot imaging	Given by sensor resolution, ∼100 × 100 px/spatial resolution dependent on optics	Given by optics, 50–100 ch./∼10 nm
Staring systems (filtering)	Given by sensor resolution, ∼Mpx	6–24 ch./∼100 nm (filter wheel) variable/∼10–40 nm (LCTF)
Whiskbroom (point-scanning)	Given by motorized platform (both *x* and *y*)	Given by detector and spectrograph, 100–1000 ch./∼2–10 nm
Pushbroom (line-scanning)	Given by camera (*x*) and motor (*y*) linear resolution/∼Mpx	Given by detector and spectrograph, 100–1000 ch./∼2–10 nm
Rotating mirror scanner	Given by camera (*x*) and motor (*y*) angular resolution/∼Mpx	Given by detector and spectrograph, 100–1000 ch./∼2–10 nm

**Table 2 sensors-19-01692-t002:** Component specifications for the optical subsystem.

Specification	Value
Wavelength range	400–1000 nm
Sensor spectral pixels	1088
Spectrograph slit width	30 μm
Spectrograph spectral resolution	2.88 nm
Focusing optics	5–50 mm, f/1.3, var. aperture and zoom
Field of View (min)	5 cm (aprox.)
Depth of Field	variable, max. @ 20 cm
Focus range	20 mm–∞

**Table 3 sensors-19-01692-t003:** Imaging device properties.

Specification	Value
Exposure range	21 μs–153 s
Signal to Noise Ratio (SNR)	up to 30 dB
Camera Gain	variable, 0–24 dB
Max frame rate at full resolution	49.5 FPS

**Table 4 sensors-19-01692-t004:** Component specifications for the actuator subsystem.

Specification	Value
Stepper driver	Pololu DRV8825 (fast decay mode)
Stepper motor	NEMA 17 standard, 12 VDC
Controller	Arduino Uno
Communications	USB 2.0, serial port
Control library	AccelStepper library
Steps per mirror revolution	19,200

**Table 5 sensors-19-01692-t005:** Communications, interface and storage specifications.

Specification	Value
WAN Interface	Gigabit Ethernet
API Protocol	RESTful (HTTP)
API Metalanguage	JSON
SSD Storage	128 GB
Main computer	Odroid XU4
Operating system	Ubuntu Mate 16.04.5 LTS
Control software language	Python 3.5.2
Measurement file format	mat or pkl
Display/touchscreen	Odroid VU7
Display resolution	800 × 480

**Table 6 sensors-19-01692-t006:** Ideal throughput/efficiency summary with the proposed device (other entries extracted from [23]). We have shown that ideal efficiencies diverge from real ones by various timing issues that should be overcome. These systems, then, can benefit from the evaluation approaches in this article.

Imaging Modality	Ideal Acquisition Time
Snapshot imaging	*T* _exp_
Staring systems	(*T*_exp_ + *T*_change_) × *N*_filters_
Whiskbroom (point-scanning)	(*T*_exp_ + *T*_move_) × *N_x_* × *N_y_*
Pushbroom (line-scanning)	(*T*_exp_ + *T*_move_) × *N*_line_
Rotating mirror scanner	(*T*_exp_ + *T*_act_) × *N*_line_

*T*_exp_ := exposure time (in s), *N*_filters_ := number of filters, *N_x_*, *N_y_* := number of spatial positions in the point-scanner, *N*_line_ := number of line positions in pushbroom and scanning systems, *T*_change_ := time needed for changing filters, *T*_move_ := time needed for moving the imaging platform or device, *T*_act_ := scanning mirror actuator rotation time.

**Table 7 sensors-19-01692-t007:** Specifications of the custom-built imaging instrument.

Specification	Value
Wavelength range	400–1000 nm
Spectral resolution	200 (5 px, 3 nm per channel)
Exposure range	10–1000 μs
Resolution (max.)	1200 × 1200 (1.44 MP)
Steps per mirror revolution	19,200
Rotation range	*π*/8 rad (45°)
Focusing optics	5–50 mm, f/1.3, var. aperture and zoom
Field of View (min)	7 × 7 cm (aprox.)
Depth of Field	variable, max. @ 20 cm
Signal to Noise Ratio (SNR)	up to 30 dB
Camera Gain	variable, 0–25 dB
Intensity distortions due to microstepping	<10% per pixel
Efficiency @ 8 bits, 100 ms	75%
USAF 1951 best resolved distance	0.5 mm at 35 cm distance

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
