# Peer review of "Custom Scanning Hyperspectral Imaging System for Biomedical Applications: Modeling, Benchmarking, and Specifications"

_sensors, 2019, doi:10.3390/s19071692_

Round 1
Reviewer 1 Report
Manuscript: sensors-460719
Title: Custom scanning hyperspectral imaging system for biomedical applications: modeling, benchmarking, and specifications
Authors: José A. Gutiérrez-Gutiérrez, Arturo Pardo *, Eusebio Real, José M. López-Higuera, Olga M. Conde
In the manuscript “Custom scanning hyperspectral imaging system for biomedical applications: modeling, benchmarking, and specifications” Gutiérrez-Gutiérrez et al. propose a detailed description of a custom-made scanning hyperspectral camera. In a first part the authors describe their approach and the components used, and in a second part propose a series of tests to assess the performance of the camera system. Overall this manuscript is well written and should be of interest to the readership.
This reviewer however has a few concerns:
Introduction:
1- First, it would be appropriate to include in the introduction a short description of other multispectral & hyperspectral imaging methods. This would help the readership to better understand the pros and cons of the proposed approach.
Materials and Methods:
2- Regarding the components used in the system, the authors should better describe the rationale behind their choices. In particular, what were the most critical choices and the tradeoffs that should be considered in designing and building such an imaging system?
3- This spectral resolution of the spectrograph is specified, at best, at 2.8nm with a telecentric input. Spectral resolution in Table 2 & 5 is shown at 1.88nm/px. Can the author provide explanations? What slit size is used?
4- Similarly, what objective lens was used (part number)? This reviewer could not find a Computar Optics Group C-mount lens for 2/3” sensor with 50mm focal and 1.4 aperture. What is the maximum resolution for this lens?
5- The authors state : “Up to 2.4MP (2048 x 1200 px) image resolution can be achieved if necessary, although lower resolutions are used in practice” What resolution is used in practice? Why limiting the spectral axis to 1200 pixels?
6- The authors performed imaging at 8 bit dynamic range. 8 bit seems a low dynamic range for biomedical optics applications. Did the author try 12 bit dynamic range (the camera looks like it can do 12 bits)?
Results:
7- This is the main concern of this reviewer: The total number of pixels used on the imaging sensor does not dictate the resolution (either spatially or spectrally). A spatial and a spectral resolution test should be performed to properly assess the system’s performance.
8- In general the system efficiency here will be mostly limited by optical design. It might be interesting to investigate the effect of a) f-number of the objective lens and b) slit size of the spectrograph.
9- Please remove mentions to “CCD”. Since the imager used in this study has a CMOS sensor, it is confusing for the readership. Maybe remove mention of either CCD or CMOS and replace by “sensor”?
10- Figure 6 & 9: Why theoretical model is so far off compared to actual measurements? Could the model be improved? What does the current theoretical model brings in terms of information?
11- Very importantly, the influence of the light source used in combination with the camera is not discussed. Without properly controlling the source, how do the authors guarantee a good SNR over the entire spectral range with a single exposure time?
12- Why are SFDI and SSOP measurement mentioned? It seems that they are performed at a single wavelength?
13- Figure 10: it seems that lines are still noticeable in the image. Why are such lines present and how it can be improved?
Author Response
Dear Reviewer 1,
Please find enclosed our manuscript revision. The PDF file includes:
A response to both reviewers,
a redlined version of the article with all corrections made visible, and
the final appearance of the manuscript.
Please, let us know if there is anything else we can do.
Yours sincerely,
Arturo Pardo

Reviewer 2 Report
Major Changes
This paper proposes a rotating-mirror scanning hyperspectral imaging device. The work of this paper is practical and logical. However, there are some problems to be further improved as well:
1. The method of this paper is not innovative enough. In fact, most of the work is done by combining other people’s methods. The design of the rotating-mirror scanning hyperspectral imaging device is simply to select the necessary equipment and combine them. In addition, in the experimental part, the rotating-mirror scanning hyperspectral imaging device was simply tested, and no new method was proposed. Authors need to highlight their innovative contributions.
2. In the introduction section, the authors highlight that scanning imaging devices will output image artifacts when the subject under measurement is in motion, but there is no further discussion in the text, just an unclear conclusion is given in the “Benchmark results and characterized equipment” section.
3. Figure 2 (A): “SCANNING MIRROR” arrow indicates inaccuracy.
Figure 5 shows that “Total time”<”< span=""> Exposure time”, please determine if the horizontal and ordinate units are correct.
4. Equation (1) error, the correct form should be .
Please explain the meaning of each special variable in Equation 15.
5. In the previous article, the author proposed Ttx, Texp etc. to represent time delays, but in the experimental part, the total time spent in each line is a combination of exposure time Texp, dynamic library time Tapi, memory buffer copying Tmcpy and actuator delays Tact, without the use of Ttx, please describe the relationship between them to ensure the article is clear.
Author Response
Dear Reviewer 2,
Please find enclosed our manuscript revision. The PDF file includes:
A response to both reviewers,
a redlined version of the article with all corrections made visible, and
the final appearance of the manuscript.
Please, let us know if there is anything else we can do.
Yours sincerely,
Arturo Pardo

Round 2
Reviewer 1 Report
Following the previous version, many improvements have greatly improved the quality of this manuscript.
This reviewer has no further comments to make on this manuscript.
Reviewer 2 Report
All comments have been responsed correctly.